Combining active learning suggestions

Tran Alasdair 1 2 alasdair.tran@anu.edu.au
http://orcid.org/0000-0002-2302-9733 Ong Cheng Soon 1 3
http://orcid.org/0000-0002-4569-016X Wolf Christian 4 5
1 Research School of Computer Science, Australian National University , Canberra, ACT , Australia
2 Data to Decisions Cooperative Research Centre , Adelaide, SA , Australia
3 Machine Learning Research Group, Data61, CSIRO , Canberra, ACT , Australia
4 Research School of Astronomy and Astrophysics, Australian National University , Canberra, ACT , Australia
5 ARC Centre of Excellence for All-sky Astrophysics (CAASTRO) , Sydney, NSW , Australia
Ventura Sebastian
Electronic publication date: 2018 Jul 23
Publication date: 2018
Volume: 4
Electronic Location ID: e157
Received 2018 Jan 5; Accepted 2018 Jun 14
Copyright: © 2018 Tran et al.
Copyright year: 2018
Copyright holder: Tran et al.
License: This is an open access article distributed under the terms of the Creative Commons Attribution License, which permits unrestricted use, distribution, reproduction and adaptation in any medium and for any purpose provided that it is properly attributed. For attribution, the original author(s), title, publication source (PeerJ Computer Science) and either DOI or URL of the article must be cited.
License URL: https://creativecommons.org/licenses/by/4.0/

Keywords: Active learning, Bandit, Rank aggregation, Benchmark, Multiclass classification

Funding: Australian Commonwealth Government’s Cooperative Research Centers Programme Australian Research Council Centre of Excellence for All-sky Astrophysics (CAASTRO) CE110001020 The research was supported by the Data to Decisions Cooperative Research Centre whose activities are funded by the Australian Commonwealth Government’s Cooperative Research Centres Programme. This research was supported by the Australian Research Council Centre of Excellence for All-sky Astrophysics (CAASTRO), through project number CE110001020. The SDSS dataset was extracted from Data Release 12 of SDSS-III. Funding for SDSS-III has been provided by the Alfred P. Sloan Foundation, the Participating Institutions, the National Science Foundation, and the U.S. Department of Energy Office of Science. The SDSS-III web site is http://www.sdss3.org/. SDSS-III is managed by the Astrophysical Research Consortium for the Participating Institutions of the SDSS-III Collaboration including the University of Arizona, the Brazilian Participation Group, Brookhaven National Laboratory, Carnegie Mellon University, University of Florida, the French Participation Group, the German Participation Group, Harvard University, the Instituto de Astrofisica de Canarias, the Michigan State/Notre Dame/JINA Participation Group, Johns Hopkins University, Lawrence Berkeley National Laboratory, Max Planck Institute for Astrophysics, Max Planck Institute for Extraterrestrial Physics, New Mexico State University, New York University, Ohio State University, Pennsylvania State University, University of Portsmouth, Princeton University, the Spanish Participation Group, University of Tokyo, University of Utah, Vanderbilt University, University of Virginia, University of Washington, and Yale University. The funders had no role in study design, data collection and analysis, decision to publish, or preparation of the manuscript.

==============================
We study the problem of combining active learning suggestions to identify informative training examples by empirically comparing methods on benchmark datasets. Many active learning heuristics for classification problems have been proposed to help us pick which instance to annotate next. But what is the optimal heuristic for a particular source of data? Motivated by the success of methods that combine predictors, we combine active learners with bandit algorithms and rank aggregation methods. We demonstrate that a combination of active learners outperforms passive learning in large benchmark datasets and removes the need to pick a particular active learner a priori. We discuss challenges to finding good rewards for bandit approaches and show that rank aggregation performs well.

Introduction

Recent advances in sensors and scientific instruments have led to an increasing use of machine learning techniques to manage the data deluge. Supervised learning has become a widely used paradigm in many big data applications. This relies on building a training set of labeled examples, which is time-consuming as it requires manual annotation from human experts.

The most common approach to producing a training set is passive learning, where we randomly select an instance from a large pool of unlabeled data to annotate, and we continue doing this until the training set reaches a certain size or until the classifier makes sufficiently good predictions. Depending on how the underlying data is distributed, this process can be quite inefficient. Alternatively we can exploit the current set of labeled data to identify more informative unlabeled examples to annotate. For instance we can pick examples near the decision boundary of the classifier, where the class probability estimates are uncertain (i.e., we are still unsure which class the example belongs to).

Many active learning heuristics have been developed to reduce the labeling bottleneck without sacrificing the classifier performance. These heuristics actively choose the most informative examples to be labeled based on the predicted class probabilities. “Overview of Active Learning” describes two families of algorithms in detail: uncertainty sampling and version space reduction.

In this paper, we present a survey of how we can combine suggestions from various active learning heuristics. In supervised learning, combining predictors is a well-studied problem. Many techniques such as AdaBoost (Freund & Schapire, 1996) (which averages predictions from a set of models) and decision trees (Breiman et al., 1984) (which select one model for making predictions in each region of the input space) have been shown to perform better than just using a single model. Inspired by this success, we propose to combine active learning suggestions with bandit and rank aggregation methods in “Combining Suggestions.”

The use of bandit algorithms to combine active learners has been studied before (Baram, El-Yaniv & Luz, 2004; Hsu & Lin, 2015). Borda count, a simple rank aggregation method, has been used in the context of multi-task learning for linguistic annotations (Reichart et al., 2008), where we have one active learner selecting examples to improve the performance of multiple related tasks (e.g., part-of-speech tagging and name entity recognition). Borda count has also been used in multi-label learning (Reyes, Morell & Ventura, 2018) to combine uncertainty information from multiple labels. As far as we know, other aggregation methods have not been explored and our work is the first time that social choice theory is used to rank and aggregate suggestions from multiple active learners.

This paper makes the following two main contributions: We empirically compare four bandit and three rank aggregation algorithms in the context of combining active learning heuristics. We apply these algorithms to 11 benchmark datasets from the UCI Machine Learning Repository (Lichman, 2013) and a large dataset from the Sloan Digital Sky Survey (SDSS) (Alam et al., 2015). The experimental setup and discussion are described in “Experimental Protocol, Results, and Discussion.”

We propose two metrics for evaluation: the mean posterior balanced accuracy (MPBA) and the strength of an algorithm. The MPBA extends the metric proposed in Brodersen et al. (2010) from the binary to the multi-class setting. This is an accuracy measure that takes class imbalance into account. The strength measure is a variation on the deficiency measure used in Baram, El-Yaniv & Luz (2004) which evaluates the performance of an active learner or combiner, relative to passive learning. The main difference between our measure and that of Baram, El-Yaniv & Luz (2004) is that ours assigns a higher number for better active learning methods and ensures that it is upper-bounded by 1 for easier comparison across datasets.

Overview of Active Learning

In this paper we consider the binary and multiclass classification settings where we would like to learn a classifier h, which is a function that maps some feature space X⊆ℝd to a probability distribution over a finite label space 𝒴: (1) h:X→p(Y)

In other words, we require that the classifier produces class probability estimates for each unlabeled example. For instance, in logistic regression with only two classes, i.e., Y={0,1}, we can model the probability that an object with feature vector x belongs to the positive class with(2) h(x;θ)=ℙ(y=1|x;θ)=11+e−θTx

and the optimal weight vector θ is learned in training. We can further consider kernel logistic regression, where the feature space X is the feature space corresponding to a given kernel, allowing for non-linear decision functions.

In active learning, we use the class probability estimates from a trained classifier to estimate a score of informativeness for each unlabeled example. In pool-based active learning, where we select an object from a pool of unlabeled examples at each time step, we require that some objects have already been labeled. In practice this normally means that we label a small random sample at the beginning. These become the labeled training set ℒT⊂X×Y and the rest form the unlabeled set U⊂X.

Now consider the problem of choosing the next example in U for querying. Labeling can be a very expensive task, because it requires using expensive equipment or human experts to manually examine each object. Thus, we want to be smart in choosing the next example. This motivates us to come up with a rule s(x; h) that gives each unlabeled example a score based only on their feature vector x and the current classifier h. Recall that the classifier produces p(𝒴), a probability estimate for each class. We use these probability estimates from the classifier over the unlabeled examples to calculate the scores: (3) s:p(Y)→ℝ

The value of s(x; h) indicates the informativeness of example x, where bigger is better. We would then label the example with the largest value of s(x; h). This will be our active learning rule r: (4) r(U;h)=argmaxx∈Us(x;h)

Algorithm 1 outlines the standard pool-based active learning setting.

Algorithm 1 The pool-based active learning algorithm.

Input: unlabeled set 𝒰, labeled training set ℒT, classifier h(x), and active learner r(U; h).	
repeat	
 Select the most informative candidate x* from 𝒰 using the active learning rule r(U; h).	
 Ask the expert to label x*. Call the label y*.	
 Add the newly labeled example to the training set: ℒT←ℒT∪{(x*,y*)}.	
 Remove the newly labeled example from the unlabeled set: U←U\{x*}.	
 Retrain the classifier h(x) using ℒT.	
until we have enough training examples.	

Coming up with an optimal rule is itself a difficult problem, but there have been many attempts to derive good heuristics. Five common ones, which we shall use in our experiments, are described in “Uncertainty Sampling” and “Version Space Reduction.”

There are also heuristics that involve minimizing the variance or maximizing the classifier certainty of the model (Schein & Ungar, 2007), but they are computationally expensive. For example, in the variance minimization heuristic, the score of a candidate example is the expected reduction in the model variance if that example were in the training set. To compute this reduction, we first need to give the example each of the possible labels, add it to the training set, and update the classifier. This is expensive to run since in each iteration, the classifier needs to be retrained k × U times, where k is the number of classes and U is the size of the unlabeled pool. There are techniques to speed this up such as using online training or assigning a score to only a small subset of the unlabeled pool. Preliminary experiments showed that these heuristics do not perform as well as the simpler ones (Tran, 2015), so we do not consider them in this paper.

A more comprehensive treatment of these active learning heuristics can be found in (Settles, 2012).

Uncertainty sampling

Lewis & Gale (1994) introduced uncertainty sampling, where we select the instance whose class membership the classifier is least certain about. These tend to be points that are near the decision boundary of the classifier. Perhaps the simplest way to quantify uncertainty is the least confidence heuristic (Culotta & McCallum, 2005), where we pick the candidate whose most likely label the classifier is most uncertain about: (5) rLC(U;h)=argmaxx∈U{−maxy∈Y p(y|x;h)}

where p(y|x; h) is the probability that the object with feature vector x belongs to class y under classifier h. For consistency, we have flipped the sign of the score function so that the candidate with the highest score is picked.

A second option is to calculate the entropy (Shannon, 1948), which measures the amount of information needed to encode a distribution. Intuitively, the closer the class probabilities of an object are to a uniform distribution, the higher its entropy will be. This gives us the heuristic of picking the candidate with the highest entropy of the distribution over the classes: (6) rHE(U;h)=argmaxx∈U{−∑y∈Yp(y|x;h)log[p(y|x;h)]}

As a third option we can pick the candidate with the smallest margin, which is defined as the difference between the two highest class probabilities (Scheffer, Decomain & Wrobel, 2001): (7) rSM(U;h)=argmaxx∈U{−(maxy∈Y p(y|x;h)−maxz∈Y\{y*}p(z|x;h))}

where y*=argmaxy∈Y p(z|x;h) and we again flip the sign of the score function. Since the sum of all probabilities must be 1, the smaller the margin is, the harder it is to differentiate between the two most likely labels.

An extension to the above three heuristics is to weight the score with the information density so that we give more importance to instances in regions of high density: (8) sID(U;h)=(1U∑k=1Esim(x,x(k)))s(x;h)

where h is the classifier, s(x; h) is the original score function of the instance with feature vector x, U is the size of the unlabeled pool, and sim(x, x(k)) is the similarity between x and another instance x(k) using the Gaussian kernel with parameter γ: (9) sim(x,x(k))=exp(γ‖x−x(k)‖2)

The information density weighting was proposed by Settles & Craven (2008) to discourage the active learner from picking outliers. Although the class membership of outliers might be uncertain, knowing their labels would probably not affect the classifier performance on the data as a whole.

Version space reduction

Instead of focusing on the uncertainty of individual predictions, we could instead try to constrain the size of the version space, thus allowing the search for the optimal classifier to be more precise. The version space is defined as the set of all possible classifiers that are consistent with the current training set. To quantify the size of this space, we can train a committee of B classifiers, ℬ = {h1, h2, …, hB}, and measure the disagreement among the members of the committee about an object’s class membership. Ideally, each member should be as different from the others as possible but still be in the version space (Melville & Mooney, 2004). In order to have this diversity, we give each member only a subset of the training examples. Since there might not be enough training data, we need to use bootstrapping and select samples with replacement. Hence this method is often called Query by Bagging (QBB).

One way to measure the level of disagreement is to calculate the margin using the class probabilities estimated by the committee (Melville & Mooney, 2004): (10) rQBBM(U;h)=argmaxx∈U{−(maxy∈Y p(y|x;ℬ)−maxz∈Y\{y*}p(z|x;ℬ))}

where(11) y*=argmaxy∈Yp(z|x;B)

(12) p(z|x;ℬ)=1B∑b∈ℬp(y|x;hb)

This looks similar to one of the uncertainty sampling heuristics, except now we use p(y|x; ℬ) instead of p(y|x; h). That is, we first average out the class probabilities predicted by the members before minimizing the margin. McCallum & Nigam (1998) offered an alternative disagreement measure which involves picking the candidate with the largest mean Kullback–Leibler (KL) divergence from the average: (13) rQBBKL(U;h)=argmaxx∈U{1B∑b=1BDKL(pb∥pℬ)}

where DKL(pb‖pB) is the KL divergence from pB (the probability distribution that is averaged across the committee B), to pb (the distribution predicted by a member b ∈ B): (14) DKL(pb∥pℬ)=∑y∈Yp(y|x;hb)  lnp(y|x;hb)p(y|x;ℬ)

For convenience, we summarize the five heuristics discussed above in Table 1.

Table 1 Summary of active learning heuristics used in our experiments.

Abbreviation	Heuristic	Objective function	
confidence	Least confidence	argmaxx∈U{−maxy∈Y p(y|x;h)}	
entropy	Highest entropy	argmaxx∈U{−∑y∈Y p(y|x;h)log[p(y|x;h)]}	
margin	Smallest margin	argmaxx∈U{−(maxy∈Y p(y|x;h)−maxz∈Y\{y*} p(z|x;h))}	
qbb-margin	Smallest QBB margin	argmaxx∈U{−(maxy∈Y p(y|x;ℬ)−maxz∈Y\{y*} p(z|x;ℬ))}	
qbb-kl	Largest QBB KL	argmaxx∈U{1B∑b=1BDKL(pb∥pℬ)}	
Note:

Notations: p(y|x; h) is the probability of that an object with feature vector x has label y under classifier h, ℬ is the set of ℬ classifiers {h1, h2, …, hB}, 𝒴 is the set of possible labels, y* is the most certain label, 𝒰 is the set of unlabeled instances, DKL(p‖q) is the Kullback–Leibler divergence of p from q, and pℬ is the class distribution averaged across classifiers in ℬ. For consistency, with heuristics that use minimization, we flip the sign of the score so that we can always take the argmax to get the best candidate.

Combining Suggestions

Out of the five heuristics discussed, which one should we use in practice when we would like to apply active learning to a particular problem? There have been some attempts in the literature to do a theoretical analysis of their performance. Proofs are however scarce, and when there is one available, they normally only work under restrictive assumptions. For example, Freund et al. (1997) showed that the query by committee algorithm (a slight variant of our two QBB heuristics) guarantees an exponential decrease in the prediction error with the training size, but only when there is no noise. In general, whether any of these heuristics is guaranteed to beat passive learning is still an open question.

Even though we do not know which one is the best, we can still combine suggestions from all of the heuristics. This can be thought of as the problem of prediction with expert advice, where each expert is an active learning heuristic. In this paper we explore two different approaches: we can either consider the advice of only one expert at each time step (with bandit algorithms), or we can aggregate the advice of all the experts (with social choice theory).

Combining suggestions with bandit theory

First let us turn our attention to the multi-armed bandit problem in probability theory (Berry & Fristedt, 1985). The colorful name originates from the situation where a gambler stands in front of a slot machine with R levers. When pulled, each lever gives out a reward according to some unknown distribution. The goal of the game is to come up with a strategy that can maximize the gambler’s lifetime rewards. In the context of active learning, each lever is a heuristic with a different ability to identify the candidate whose labeling information is most valuable.

The main problem in multi-armed bandits is the trade-off between exploring random heuristics and exploiting the best heuristic so far. There are many situations in which we find our previously held beliefs to be completely wrong. By always exploiting, we could miss out on the best heuristic. On the other hand, if we explore too much, it could take us a long time to reach the desired accuracy.

Bandit algorithms do not need to know the internal workings of the heuristics, but only the reward received from using any of them. At each time step, we receive a reward from a heuristic, and based on the history of all the rewards, the bandit algorithm can decide on which heuristic to pick next. Formally, we need to learn the function(15) b:(JR×[0,1])n→JR

where b is the bandit algorithm, the reward is normalized between 0 and 1, Jℛ is the index set over the set of heuristics ℛ, and n is the time horizon.

What would be an appropriate reward w in this setting? We propose using the incremental increase in the performance of the test set after the candidate is added to the training set. This, of course, means that we need to keep a separate labeled test set around, just for the purpose of computing the rewards. We could, as is common practice in machine learning, use cross validation or bootstrap on ℒT to estimate the generalization performance. However for simplicity of presentation we use a separate test set ℒS.

Figure 1 and Algorithm 2 outline how bandits can be used in pool-based active learning. The only difference between the bandit algorithms lies in the Select function that selects which heuristic to use, and the Update function that updates the algorithm’s selection parameters when receiving a new reward.

Figure 1 Active learning pipeline with bandit algorithms.

We need to collect rewards w from the test set ℒS in order to decide which heuristic to choose at each time step. This routine is indicated by the red arrows. Notations: ℛ is the set of heuristics {r1, …, rR}, ℒT is the training set, ℒS is the test set, 𝒰 is the unlabeled set, and p(𝒴) is the predicted class probabilities on the unlabeled data 𝒰.

There have been some attempts to combine active learning suggestions in the literature. Baram, El-Yaniv & Luz (2004) used the EXP4 multi-armed bandit algorithm to automate the selection process. They proposed a reward called the classification entropy maximization, which can be shown to grow at a similar rate to the true accuracy in binary classification with support vector machines (SVMs). We will not compare our results directly with those in Baram, El-Yaniv & Luz (2004) since we would like to evaluate algorithms that can work with both binary and multi-class classification. Our experiments also use logistic regression which produces probability estimates directly, rather than SVMs which can only produce unnormalized scores. Hsu & Lin (2015) studied an improved version of EXP4, called EXP4.P, and used importance weighting to estimate the true classifier performance using only the training set. In this paper, we empirically compare the following four bandit algorithms: Thompson sampling, OC-UCB, kl-UCB, and EXP3++.

Algorithm 2 Pool-based active learning with bandit theory. Note that in addition to the set of active learning heuristics ℛ and the test set ℒS, some bandit algorithms also need to know n, the maximum size of the training set, in advance.

Input: unlabeled set 𝒰, labeled training set ℒT, labeled test set ℒS, classifier h, desired training size n, set of active learning heuristics ℛ, and bandit algorithm b with two functions Select and Update.	
while |ℒT| < n do	
  Select a heuristic r* ∈ ℛ according to Select.	
  Select the most informative candidate x* from 𝒰 using the chosen heuristic r* (𝒰; h).	
  Ask the expert to label x*. Call the label y*.	
  Add the newly labeled example to the training set: ℒT←ℒT∪{(x*,y*)}.	
  Remove the newly labeled example from the unlabeled set: U←U\{x*}.	
  Retrain the classifier h(x) using ℒT.	
  Run the updated classifier on the test set ℒS to compute the increase in the performance w.	
  Update the parameters of b with Update(w).	
end	

Thompson sampling

The oldest bandit algorithm is Thompson sampling (Thompson, 1933) which solves the exploration-exploitation trade-off from a Bayesian perspective.

Let Wi be the reward of heuristic ri ∈ ℛ. Observe that even with the best heuristic, we still might not score perfectly due to having a poor classifier trained on finite data. Conversely, a bad heuristic might be able to pick an informative candidate due to pure luck. Thus there is always a certain level of randomness in the reward received. Let us treat the reward Wi as a normally distributed random variable with mean 𝜈i and variance τi2: (16) (Wi|νi)∼N(νi,τi2)

If we knew both 𝜈i and τi2 for all heuristics, the problem would become trivially easy since we just need to always use the heuristic that has the highest mean reward. In practice, we do not know the true mean of the reward 𝜈i, so let us add a second layer of randomness and assume that the mean itself follows a normal distribution: (17) νi∼N(μi,σi2)

To make the problem tractable, let us assume that the variance τi2 in the first layer is a known constant. The goal now is to find a good algorithm that can estimate μi and σi2.

We start with a prior on μi and σi2 for each heuristic ri. The choice of prior does not usually matter in the long run. Since initially we do not have any information about the performance of each heuristic, the appropriate prior value for μi is 0, i.e., there is no evidence (yet) that any of the heuristics offers an improvement to the performance.

In each round, we draw a random sample 𝜈i′ from the normal distribution N(μi,σi2) for each i and select heuristic r* that has the highest sampled value of the mean reward: (18) r*=arg maxi v′i

We then use this heuristic to select the object that is deemed to be the most informative, add it to the training set, and retrain the classifier. Next we use the updated classifier to predict the labels of objects in the test set. Let w be the reward observed. We now have a new piece of information that we can use to update our prior belief about the mean μ* and the variance σ*2 of the mean reward. Using Bayes’ theorem, we can show that the posterior distribution of the mean reward remains normal,(19) (ν*∣W*=w)~(μ′*,σ′*2)

with the following new mean and variance: (20) μ∗′=μ*τ*2+wσ*2σ*2+τ*2    σ∗′2=σ*2τ*2σ*2+τ*2

Algorithm 3 summarizes the Select and Update functions used in Thompson sampling.

Algorithm 3 Thompson sampling with normally distributed rewards. Notations: ℛ is the set of R heuristics, μ is the mean parameter of the average reward, σ2 is the variance parameter of the average reward, τ2 is the known variance parameter of the reward, and w is the actual reward received.

function Select()	
   for i ∈ {1, 2, …, R} do	
     𝜈i′ draw a sample from N(μi,σi2)	
   end	
   Select the heuristic with the highest sampled value: r*←argmaxi ν′i	
function Update(w)	
   μ*←μ*τ*2+wσ*2σ*2+τ*2   σ*2←σ*2τ*2σ*2+τ*2	

Upper confidence bounds

Next we consider the Upper Confidence Bound (UCB) algorithms which use the principle of “optimism in the face of uncertainty.” In choosing which heuristic to use, we first estimate the upper bound of the reward (that is, we make an optimistic guess) and pick the one with the highest bound. If our guess turns out to be wrong, the upper bound of the chosen heuristic will decrease, making it less likely to get selected in the next iteration.

There are many different algorithms in the UCB family, e.g., UCB1-TUNED & UCB2 (Auer, Cesa-Bianchi & Fischer, 2002a), V-UCB (Audibert, Munos & Szepesvári, 2009), OC-UCB (Lattimore, 2015), and kl-UCB (Cappé et al., 2013). They differ only in the way the upper bound is calculated. In this paper, we only consider the last two. In Optimally Confident UCB (OC-UCB), Lattimore (2015) suggests that we pick the heuristic that maximizes the following upper bound: (21) r*=argmaxi(wi¯+αTi(t)ln(ψnt))

where wi¯ is the average of the rewards from ri that we have observed so far, t is the time step, Ti(t) is the number times we have selected heuristic ri before step t, and n is the maximum number of steps that we are going to take. There are two tunable parameters, α and ψ, which the author suggests setting to 3 and 2, respectively.

In kl-UCB, Cappé et al. (2013) suggest that we can instead consider the KL-divergence between the distribution of the current estimated reward and that of the upper bound. In the case of normally distributed rewards with known variance σ2, the chosen heuristic would be (22) r*=argmaxi(wi¯+2σ2ln(Ti(t))t)

Algorithms 4 and 5 summarize these two UCB approaches. Note that the size of the reward w is not used in Update (w) of UCB, except to select the best arm.

Algorithm 4 Optimally Confident UCB. Notations: n is the time horizon (maximum number of time steps), t is the current time step, Ti(t) counts how many times heuristic i has been selected before step t, w is the reward received, and wi¯ is the average of the rewards from ri so far.

fuction Select()	
  r*←argmaxiwi¯+3Ti(t)ln(2nt)	
function Update(w)	
  t←t+1	
  T*(t)←T*(t−1)+1	

Algorithm 5 kl-UCB with normally distributed rewards. Notations: σ2 is the variance of the rewards, t is the current time step, Ti(t) counts how many times heuristic i has been selected before step t, w is the reward received, and wi¯ is the average of the rewards from ri so far.

function Select()	
  r*←argmaxiwi¯+2σ2ln(Ti(t))t	
function Update(w)	
  t←t+1	
  T*(t)←T*(t−1)+1	

EXP3++

The exponential-weight algorithm for exploration and exploitation (EXP3) was first developed by Auer et al. (2002b) to solve the non-stochastic bandit problem where we make no statistical assumptions about the reward distribution. This is also often known as the adversarial setting, where we have an adversary who generates an arbitrary sequence of rewards for each heuristic in advance. Like Thompson sampling, the algorithm samples from a probability distribution at each step to pick a heuristic. Here however, we construct the distribution with exponential weighting (hence the name EXP3). We shall test Seldin & Slivkins (2014)’s EXP3++ algorithm (see Algorithm 6). This is a generalization of the original EXP3 and it has been shown to perform well in both the stochastic (where the reward of each heuristic follows an unknown reward distribution) and the adversarial regime.

Algorithm 6 EXP3++ algorithm. Notations: ℛ is the set of R heuristics, t is the current time step, β is a parameter used to weight the heuristics for selection, ξi and εi are used to compute the loss Li, ρ is the distribution from which a heuristic is sampled, and w is the reward received.

function Select()	
  β=12lnRtR	
  for i ∈ {1, 2, …, R} do	
   ξi=18 In(t)2tmin(1,1t(Li−min(L)))2	
   ξi=min(12R,β,ξi)	
   ρi=e−β*Li∑je−β*Lj	
  end	
  r*← draw a sample from ℛ with probability distribution ρ.	
function Update (w)	
  t←t+1	
  T*(t)←T*(t−1)+1	
  L*←L*+(1−w)(1−∑jεj)W*+ε*	

Combining suggestions with social choice theory

A drawback of the above bandit methods is that at each iteration, we could only use one suggestion from one particular heuristic. EXP4 and EXP4.P algorithms can take advice from all heuristics by maintaining a weight on each of them. However, being a bandit method, they require designing a reward scheme. If the reward is the performance on a test set, we would need to keep around a separate subset of the data, which is expensive and sometimes impossible to obtain in practice. This leads us to social choice theory, which can combine suggestions like EXP4 and EXP4.P, while not needing the concept of a reward. Originally developed by political scientists like Nicolas de Condorcet and Jean-Charles de Borda, this field of study is concerned with how we aggregate preferences of a group of people to determine, for example, the winner in an election (List, 2013). It has the nice property that everyone (or in our context, every active learning heuristic) has a voice.

For each heuristic, we assign a score to every candidate with the score function s(x, h) like before. We are neither interested in the actual raw scores nor the candidate with the highest score. Instead, we only need a ranking of the candidates, which is achieved by a function k(s,U) that provides a ranking of the unlabeled examples according to their scores. For example, k could assign the candidate with the highest score a rank of 1, the next best candidate a rank of 2, and so on. An aggregation function c will then combine all the rankings into a combined ranking,(23) c:σ(JU)R→σ(JU)

where σ(JU) is a permutation over the index set of the unlabeled pool 𝒰 and R is the number of heuristics. From these we can pick the highest-ranked candidate to annotate. See Table 2 for an example.

Table 2 An example of how to convert raw scores into a ranking.

Score	s(x; h)	0.1	0.9	0.3	0.8	
Rank	k(s, U)	4	1	3	2	

The main difference between this approach and the bandit algorithms is that we do not consider the reward history when combining the rankings. Here each heuristic is assumed to always have an equal weight. A possible extension, which is not considered in this paper, is to use the past performance to re-weight the heuristics before aggregating at each step. Figure 2 and Algorithm 7 provide an overview of how social choice theory is used in pool-based active learning.

Figure 2 Active learning pipeline with rank aggregation methods.

Unlike the bandit pipeline, there is only one cycle in which we aggregate information from all heuristics. Additional notation: σ(JU) is a permutation (i.e., rank) on the index set of the unlabeled data.

Algorithm 7 Pool-based active learning with social choice theory.

Input: unlabeled set 𝒰, labeled training set ℒT, classifier h, set of active learning suggestions ℛ, ranking function k, and rank aggregator c.	
repeat:	
  for r ∈ ℛ do	
   Rank all the candidates in 𝒰 with k.	
  end	
  Aggregate all the rankings into one ranking using the aggregator c.	
  Select the highest-ranked candidate x* from 𝒰.	
  Ask the expert to label x*. Call the label y*.	
  Add the newly labeled example to the training set: ℒT←ℒT∪{(x*,y*)}.	
  Remove the newly labeled example from the unlabeled set: U←U\{x*}.	
  Retrain the classifier h(x) using ℒT.	
until we have enough training examples.	

The central question in social choice theory is how we can come up with a good preference aggregation rule. We shall examine three aggregation rules: Borda count, the geometric mean, and the Schulze method.

In the simplest approach, Borda count, we assign an integer point to each candidate. The lowest-ranked candidate receives a point of 1, and each candidate receives one more point than the candidate below. To aggregate, we simply add up all the points each candidate receives from every heuristic. The candidate with the most points is declared the winner and is to be labeled next. We can think of Borda count, then, as ranking the candidate according to the arithmetic mean.

An alternative approach is to use the geometric mean, where instead of adding up the points, we multiply them. Bedö & Ong (2016) showed that the geometric mean maximizes the Spearman correlation between the ranks. Note that this method requires the ranks to be scaled so that they lie strictly between 0 and 1. This can be achieved by simply dividing the ranks by (U + 1), where U is the number of candidates.

The third approach we consider is the Schulze method (Schulze, 2011). Out of the three methods considered, this is the only one that fulfills the Condorcet criterion, i.e., the winner chosen by the algorithm is also the winner when compared individually with each of the other candidates. However, the Schulze method is more computationally intensive since it requires examining all pairs of candidates. First we compute the number of heuristics that prefer candidate xi to candidate xj, for all possible pairs (xi, xj). Let us call this d(xi, xj). Let us also define a path from candidate xi to xj as the sequence of candidates, {xi, x1, x2, …, xj}, that starts with xi and ends with xj, where, as we move along the path, the number of heuristics that prefer the current candidate over the next candidate must be strictly decreasing. Intuitively, the path is the rank of a subset of candidates, where xi is the highest-ranked candidate and xj is at the lowest-ranked.

Associated with each path is a strength p, which is the minimum of d(xi, xj) for all consecutive xi and xj along the path. The core part of the algorithm involves finding the path of the maximal strength from each candidate to every other. Let us call p(xi, xj) the strength of strongest path between xi and xj. Candidate xi is a potential winner if p(xi,xj)≥p(xj,xi) for all other xj. This problem has a similar flavor to the problem of finding the shortest path. In fact, the implementation uses a variant of the Floyd–Warshall algorithm to find the strongest path. This is the most efficient implementation that we know of, taking cubic time in the number of candidates.

We end this section with a small illustration of how the three aggregation algorithms work in Table 3.

Table 3 An example of how social choice theory algorithms rank candidates by aggregating three heuristics: r1, r2, and r3.

There are four candidates in the unlabeled pool: A, B, C, and D.

(a) An example of how the three heuristics rank four candidates A, B, C, and D. For instance, heuristic r1 considers B to be the highest rank candidate, followed by A, C, and D.	
Heuristic	Ranking	
r1	B A C D	
r2	A C B D	
r3	B D C A	
(b) Aggregated ranking with Borda count and geometric mean. The scores are determined by the relative ranking in each heuristic. For example, A is ranked second by r1, first by r1, and last by r3, thus giving us a score of 3, 4, and 1, respectively. In both methods, candidate B receives the highest aggregated score.	
Candidate	Borda count	Geometric mean	
A	3 + 4 + 1 = 8	3 × 4 × 1 = 12	
B	4 + 2 + 4 = 10	4 × 2 × 4 = 32	
C	2 + 3 + 2 = 7	2 × 3 × 2 = 12	
D	1 + 1 + 3 = 5	1 × 1 × 3 = 3	
(c) Aggregated ranking with the Schulze method. The table shows the strongest path strength p(xi, xj) between all pairs of candidates. For example, p(B, D) = 3 because the path B → D is the strongest path from B to D, where three heuristics prefer B over D. Candidate B is the winner since p(B, A) > p(A, B), p(B, C) > p(C, B), and p(B, D) > p(D, B).	
From/To	A	B	C	D	
A	–	1	2	2	
B	2	–	2	3	
C	1	1	–	2	
D	2	0	1	–	

Experimental Protocol

We use 11 classification datasets taken from the UCI Machine Learning Repository (https://archive.ics.uci.edu/ml/) (Lichman, 2013), with a large multiclass classification dataset which we extracted from the SDSS project (DOI 10.5281/zenodo.58500) (Alam et al., 2015). The code for the experiments can be found on our GitHub repository (https://github.com/chengsoonong/mclass-sky). Table 4 shows the size and the number of classes in each dataset, along with the proportion of the samples belonging to the majority class and the maximum achievable performance using logistic regression. These datasets were chosen such that we have an equal number of binary and multiclass datasets, and a mixture of small and large datasets.

Table 4 Overview of datasets.

Dataset	Size	No. of classes	No. of features	Majority class (%)	Max performance (MPBA) (%)	
Glass	214	6	10	33	65	
Ionosphere	351	2	34	64	89	
Iris	150	3	4	33	90	
Magic	19,020	2	11	65	84	
Miniboone	129,596	2	50	72	88	
Pageblock	5,473	5	10	90	79	
Pima	733	2	8	66	71	
SDSS	2,801,002	3	11	61	90	
Sonar	208	2	60	53	78	
Vehicle	846	4	18	26	81	
Wine	178	3	13	40	94	
WPBC	194	2	34	76	58	
Note:

The following datasets are from the UCI Machine Learning Repository: glass, ionosphere, iris, magic, miniboone, pageblock, pima, sonar, vehicle, wine, and wpbc. In particular, the vehicle dataset comes from the Turing Institute, Glasgow, Scotland. The sdss dataset was extracted from Data Release 12 of SDSS-III.

For each dataset, we use Scikit-learn (Pedregosa et al., 2011) to train a logistic regression model using a 10-fold stratified shuffled cross-validation. Here “stratified” means that the proportion of the classes remains constant in each split. We standardize all features to have zero mean and unit variance. Although all examples have already been labeled, we simulate the active learning task by assuming that certain examples do not have any labels. For each fold, the unlabeled pool size is 70% of data up to a maximum of 10,000 examples, and the test pool consists of the remaining examples up to a maximum of 20,000. We assume all test examples are labeled. We initialize the classifier by labeling 10 random instances and using them as the initial training set. The heuristics are fast enough such that we can assign a score to every unlabeled instance at every time step. We use logistic regression with a Gaussian kernel approximation and an L2 regularizer. In the binary case, the loss function is(24) L=12θTθ+C∑i=1nln(1+exp(−yi(θTf(xi))))

where xi is the feature vector of the ith example, yi ∈ {0, 1} is the label of xi, and n is the training size. The term 12θTθ is the regularization term to ensure that the weight vector θ is not too large, and C is a regularization hyperparameter in [10−6, 106] which we find using grid search. To speed up the training time while using the Gaussian kernel, we approximate the feature map of the kernel with Random Kitchen Sinks (Rahimi & Recht, 2008), transforming the raw features xi into a fixed 100-dimensional feature vector f (xi). In the multiclass case, we use the One-vs-Rest strategy, where for every class we build a binary classifier that determines whether a particular example belongs to that class or not. For the QBB algorithms, we train a committee of seven classifiers, where each member is given a sample of maximum 100 examples that have already been labeled.

For the bandit algorithms, we use the increase in the MPBA on the test set as the reward. The MPBA can be thought of as the expected value of the average recall, where we treat the recall as a random variable that follows a Beta distribution. Compared to the raw accuracy score, this metric takes into account class imbalance. This is because we first calculate the recall in each class and then take the average, thus giving each class an equal weight. Refer to Appendix A for the derivation of the MPBA, which extends Brodersen et al. (2010)’s formula from the binary to the multiclass setting.

In total, we test 17 query strategies. This includes passive learning, eight active learning heuristics, five bandit algorithms, and three aggregation methods. The bandit algorithms include the four described in “Combining Suggestions with Bandit Theory” and a baseline called explore which simply selects a random heuristic at each time step. In other words, we ignore the rewards and explore 100% of the time. For all bandit and rank aggregation methods, we take advice from six representative experts: passive, confidence, margin, entropy, qbb-margin, and qbb-kl. We have not explored how adding the heuristics with information density weighting to the bandits would impact the performance. See Table 5 for a list of abbreviations associated with the methods.

Table 5 Summary of active learning heuristics and combiners used in the experiments.

Abbreviation	Type	Description	
passive	Heuristic	Passive learning	
confidence	Heuristic	Least confidence heuristic	
w-confidence	Heuristic	Least confidence heuristic with information density weighting	
margin	Heuristic	Smallest margin heuristic	
w-margin	Heuristic	Smallest margin heuristic with information density weighting	
entropy	Heuristic	Highest entropy heuristic	
w-entropy	Heuristic	Highest entropy heuristic with information density weighting	
qbb-margin	Heuristic	Smallest QBB margin heuristic	
qbb-kl	Heuristic	Largest QBB KL-divergence heuristic	
explore	Bandit	Bandit algorithm with 100% exploration	
thompson	Bandit	Thompson sampling	
ocucb	Bandit	Optimally confidence UCB algorithm	
klucb	Bandit	kl-UCB algorithm	
exp3++	Bandit	EXP3++ algorithm	
borda	Aggregation	Aggregation with Borda count	
geometric	Aggregation	Aggregation with the geometric mean	
schulze	Aggregation	Aggregation with the Schulze method	

Given that there are 12 datasets, each with 17 learning curves, we need a measure that can summarize in one number how well a particular heuristic or policy does. Building on Baram, El-Yaniv & Luz (2004)’s deficiency measure, we define the strength of an active learner or a combiner relative to passive learning as (25) Strength(h;m)=1−∑t=1n(m(max)−m(h,t))∑t=1n(m(max)−m(passive,t))

where m is a chosen metric (e.g., accuracy rate, MPBA), m(max) is the best possible performance1, and m(h, t) is the performance achieved using the first t examples selected by heuristic h. We can think of the summation as the area between the best possible performance line and the learning curve of h. The better the heuristic is, the faster it would approach this maximum line, and thus the smaller the area. Finally, so that we can compare the performance across datasets, we normalize the measure with the area obtained from using just passive learning. Refer to Fig. 3 for a visualization of the strength measure.

Figure 3 An illustration of the MPBA strength measure.

It is proportional to the shaded area between the (red) passive learning curve and the (blue) active learning curve. The bigger the area is, the more the active learner outperforms the passive learner. The top dotted line indicates the maximum performance achieved.

We evaluate the algorithm performance with two metrics: the accuracy score and the MPBA. The accuracy score is the percentage of instances in the test set where the predicted label matches the true label. If a dataset has a dominant class, then the accuracy score of instances within that class will also dominate the overall accuracy score. The MPBA, on the other hand, puts an equal weight on each class and thus favors algorithms that can predict the label of all classes equally well.

The heuristics with information density weighting and Thompson sampling have a few additional hyperparameters. To investigate the effect of these hyperparameters, we pick one binary dataset (glass) and one multiclass dataset (ionosphere) to investigate. Both of these are small enough to allow us to iterate through many hyperparameter values quickly. With w-confidence, w-margin, and w-entropy, we set γ in the Gaussian kernel to be the inverse of the 95th percentile of all pairwise distances. This appears to work well, as shown in Fig. 4. For thompson, the prior values for μ, σ2 and the value of τ2 seem to have little effect on the final performance (see Fig. 5). We set the initial μ to 0.5, the initial σ2 to 0.02, and τ2 to 0.02.

Figure 4 Effect of γ on w-confidence and w-margin using the glass and ionosphere datasets.

We examine six different values for γ: the 50th, 60th, 70th, 90th, 95th, and 99th percentile of the pairwise L1-distance between the data points. For the glass dataset (A), changing value of γ has minimal effect on the results, while for the ionosphere dataset (B), using the 90th percentile and above seems to work well.

Figure 5 Effect of the initial values of the parameters in thompson.

We test 16 combinations of μ, σ2, and τ2 on the glass (A) and ionosphere (B) dataset. Varying these values does not seem to affect the final performance.

Results

Figures 6 and 7 show the strengths of all methods that we consider, while Figs. 8 and 9 provide selected learning curves. Plots for the six small datasets with fewer than 500 examples (glass, ionosphere, iris, sonar, wine, and wpbc) are shown in Figs. 6 and 8. Plots for the two medium-sized datasets (pima and vehicle) and the four large datasets (magic, miniboone, pageblocks, and sdss) are shown in Figs. 7 and 9. Each figure contains two subfigures, one reporting the raw accuracy score, while the other showing the MPBA score.

Figure 6 Boxplots of the accuracy and MPBA strength of the 16 active learning strategies, relative to passive learning, using the small datasets (glass, ionosphere, iris, sonar, wine, and wpbc).

The more positive the strength is, the better the heuristic/combiner is. Gray boxes represent individual heuristics; blue boxes represent bandit algorithms, and red boxes are for rank aggregation methods. A strategy that is above the zero line is better than passive learning. Each boxplot contains 10 trials. The accuracy score (A, C, E, G, I, and K) is a simple metric that simply counts up the number of correct predictions. The MPBA score (B, D, F, H, J, and L), being the weighted average of the recall and precision, gives an equal representation to each class. The boxes represent the quartiles and the whiskers extend to 1.5 times of the interquartile range.

Figure 7 Boxplots of the accuracy and MPBA strength of the 16 active learning strategies, relative to passive learning, using medium to the large datasets (magic, miniboone, pageblocks, pima, sdss, and vehicle).

The more positive the strength is, the better the heuristic/combiner is. Gray boxes represent individual heuristics; blue boxes represent bandit algorithms, and red boxes are for rank aggregation methods. A strategy that is above the zero line is better than passive learning. Each boxplot contains 10 trials. The accuracy score (A, C, E, G, I, and K) is a simple metric that simply counts up the number of correct predictions. The MPBA score (B, D, F, H, J, and L), being the weighted average of the recall and precision, gives an equal representation to each class. The boxes represent the quartiles and the whiskers extend to 1.5 times of the interquartile range.

Figure 8 Selected accuracy and MPBA learning curves for the small datasets (glass, ionosphere, iris, sonar, wine, and wpbc).

As it would get too cluttered to plot 17 learning curves, we only show the accuracy (A, C, E, G, I, and K) and MPBA (B, D, F, H, J, and L) learning curves for passive, confidence, exp3++, and borda. The learning curves are averaged over 10 trials. The dotted horizontal line shows the performance obtained from using the whole training data.

Figure 9 Selected accuracy and MPBA learning curves for the medium to large datasets (magic, miniboone, pageblocks, pima, sdss, and vehicle).

As it would get too cluttered to plot 17 learning curves, we only show the accuracy (A, C, E, G, I, and K) and MPBA (B, D, F, H, J, and K) learning curve for passive, confidence, exp3++, and borda. The learning curves are averaged over 10 trials. The dotted horizontal line shows the performance obtained from using the whole training data.

Active learning methods generally beat passive learning in four of the six small datasets—glass, ionosphere, iris, and wine. This can be seen by the fact that the boxplots are mostly above the zero line in Fig. 6. For sonar and wpbc, the results are mixed—active learning has little to no effect here. The wpbc dataset is particularly noisy—our classifier cannot achieve an MPBA score greater than 60% (see Fig. 8). Thus it is not surprising that active learning does not perform well here since there is not much to learn to begin with.

The advantage of active learning becomes more apparent with the larger datasets like magic, miniboone, pageblocks, and sdss. Here there is a visible gap between the passive learning curve and the active learning curve for most methods. For instance, using a simple heuristic such as confidence in the pageblocks dataset results in an average MPBA score of 74% after 1,000 examples, while passive learning can only achieves 67% (see Fig. 9F).

Out of the eight active learning heuristics tested, the heuristics with the information density weighting (w-confidence, w-margin, and w-entropy) generally perform worse than the ones without the weighting. qbb-kl performs the best in pageblocks while it can barely beat passive learning in other datasets. The remaining heuristics—confidence, margin, entropy, and qbb-margin—perform equally well in all datasets.

We find no difference in performance between the bandit algorithms and the rank aggregation methods. Combining active learners does not seem to hurt the performance, even if we include a poorly performing heuristic such as qbb-kl.

For bandit algorithms, it is interesting to note that thompson favors certain heuristics a lot more than others, while the behavior of exp3++, ocucb, and klucb is almost indistinguishable from explore, where we explore 100% of the time (see Fig. 10). Changing the initial values of μ, σ2, and τ2 changes the order of preference slightly, but overall, which heuristics thompson picks seems to correlate with the heuristic performance. For example, as shown in Fig. 11, passive and qbb-kl tend to get chosen less often than others in the ionosphere dataset.

Figure 10 Selection frequencies of heuristics in thompson and exp3++, with the large datasets (magic, miniboone, pageblocks, pima, sdss, and vehicle).

The plots show how often each of the heuristics gets selected over time. The selection frequencies are averaged over 10 trials. thompson (A–F) favors certain heuristics more strongly than others. In contrast, exp3++ (G–L) favors uniform exploration more, sampling each heuristic with roughly equal weights. The plots for ocucb and klucb are not shown here, but they are similar to exp3++.

Figure 11 The effect of the initial values of the parameters in thompson on the heuristic selection frequencies.

We test 16 combinations of μ, σ2, and τ2 on the glass and ionosphere dataset. Which heuristics thompson picks seems to correlate with the heuristic performance. For example, in ionosphere, passive (the dotted purple line) and qbb-kl (the dashed dark blue line) tend to get picked less often than others.

Discussion

The experimental results allow us to answer the following questions:Can active learning beat passive learning? Yes, active learning can perform much better than passive learning, especially when the unlabeled pool is large (e.g., sdss, miniboone, pageblock). When the unlabeled pool is small, the effect of active learning becomes less apparent, as there are now fewer candidates to choose from. This can be seen in Fig. 12, where we show that artificially reducing the unlabeled pool results in a reduction in the final performance. At the same time, having a small test set also makes the gap between the active learning curve and the passive learning curve smaller (see Figs. 12C and 12F). This further contributes to the poorer performance on the smaller datasets. In any case, when a dataset is small, we can label everything so active learning is usually not needed.

Can active learning degrade performance? Yes, there is no guarantee that active learning will always beat passive learning. For example, w-entropy actually slows down the learning in the many datasets. However, this only happens with certain heuristics, like those using the information density weighting.

What is the best single active learning heuristic? All of confidence, margin, entropy, and qbb-margin have a similar performance. However confidence is perhaps the simplest to compute and thus is a good default choice in practice.

What are the challenges in using bandit algorithms?

Designing a good reward scheme is difficult. This paper uses the increase in the classifier performance as the reward. However this type of reward is non-stationary (i.e., it gets smaller after each step as learning saturates) and the rewards will thus eventually go to zero.

In practice, we do not have a representative test set that can be used to compute the reward. As a workaround, Hsu & Lin (2015) computed the reward on the training set and then used importance weighting to remove any potential bias. For this to work, we need to ensure that every training example and every active learning suggestion have a non-zero probability of being selected in each step.

Finally, some bandit algorithms such as Thompson sampling assumes that the reward follows a certain distribution (e.g., Gaussian). However, this assumption is unrealistic.

What are the challenges in using rank aggregation algorithms?

We need to compute the scores from all heuristics at every time step. This might not be feasible if there are too many heuristics or if we include heuristics that require a large amount of compute power (e.g., variance minimization).

The Schulze method uses O(n2) space, where n is the number of candidates. This might lead to memory issues if we need to rank a large number of candidates from the unlabeled pool.

Before aggregating the rankings, we throw away the score magnitudes, which could cause a loss of information.

Unlike bandit algorithms, all of the rank aggregators always give each heuristic an equal weight.

Which method should I use in practice to combine active learners? Since there is no difference in performance between various combiners, we recommend using a simple rank aggregator like Borda count or geometric mean if we do not want to select a heuristic a priori. Rank aggregators do not need a notion of a reward—we simply give all suggestions an equal weight when combining. Thus we neither need to a keep a separate test set, nor do we need to worry about designing a good reward scheme.

Figure 12 Effect of the pool size on the learning curves. We pick two large datasets—pageblocks and sdss—to investigate how the size of the pool affects the performance.

(A) and (D) are the original learning curves from Figs. 9F and 9J (we only show the first 200 examples so that all figures have the same scale). For (B) and (E), we use the same test pool, but the unlabeled pool now only has a maximum of 300 candidates. Finally, for (C) and (F) the combined test pool and training pool have a size of 300.

Our investigation has a few limitations. Firstly, we empirically compare algorithms that only work with single-label classification problems. Nowadays, many problems require multi-label learning, in which each example is allowed to be in more than one class. Our methods can be extended to work with multi-label datasets with the following modifications. We first need a multi-label classifier. This can be as simple as a collection of binary classifiers, each of which produces the probability that an example belongs to a particular class. For each class, we can use an active learning heuristic to assign a score to each unlabeled example as before. However now we need to aggregate the scores among the classes. As suggested by Reyes, Morell & Ventura (2018), we can use any aggregation method like Borda count to combine these scores. In effect, the multi-label learning problem adds an extra layer of aggregation into the pipeline.

Another limitation of our methods is that our active learning methods are myopic. That is, in each iteration, we only pick one instance to give to a human expert for labeling. In many practical applications like astronomy, batch-mode active learning is preferred, as it is much more cost efficient to obtain multiple labels simultaneously. One naive extension is to simply choose the m highest ranked objects using our current methods. However, it is possible to have two unlabeled objects whose class membership we are currently uncertain about, but because they have very similar feature vectors, labeling only one of them would allow us to predict the label of the other one easily. More sophisticated batch-mode active learning approaches have been proposed to take into account other factors such as the diversity of a batch and the representativeness of each batch example. These approaches include looking at the angles between hyperplanes in support vector machines (Brinker, 2003), using cluster analysis (Xu, Akella & Zhang, 2007), and using an evolutionary algorithm (Reyes & Ventura, 2018). How to aggregate suggestions from these approaches is an interesting problem for future work.

Conclusion

In this paper we compared 16 active learning methods with passive learning. Our three main findings are: active learning is better than passive learning; combining active learners does not in general degrade the performance; and social choice theory provides more practical algorithms than bandit theory since we do not need to design a reward scheme.

Appendix A: Posterior Balanced Accuracy

Most real-world datasets are unbalanced. In the SDSS dataset, for example, there are 4.5 times as many galaxies as quasars. The problem of class imbalance is even more severe in the pageblocks dataset, where one class makes up 90% of the data and the remaining four classes only make up 10%. An easy fix is to under sample the dominant class when creating the training and test sets. This, of course, means that the size of these sets are limited by the size of the minority class.

When we do not want to alter the underlying class distribution or when larger training and test sets are desired, we need a performance measure that can correct for the class imbalance. Brodersen et al. (2010) show that the posterior balanced accuracy distribution can overcome the bias in the binary case. We now extend this idea to the multi-class setting.

Suppose we have k classes. For each class i between 1 and k, there are Ni objects in the universe. Given a classifier, we can predict the label of every object and compare our prediction with the true label. Let Gi be the number of objects in class i that are correctly predicted. Then we define the recall Ai of class i as(26) Ai=GiNi

The problem is that it is not feasible to get the actual values of Gi and Ni since that would require us to obtain the true label of every object in the universe. Thus we need a method to estimate these quantities when we only have a sample. Initially we have no information about Gi and Ni, so we can assume that each Ai follows a uniform prior distribution between 0 and 1. This is the same as a Beta distribution with shape parameters α = β = 1: (27) Ai∼Beta(1,1)

The probability density function (PDF) of Ai is then(28) fAi(a)=Γ(α+β)Γ(α)Γ(β)aα−1(1−a)β−1   ∝a1−1(1−a)1−1

where Γ(α) is the gamma function.

After we have trained the classifier, suppose we have a test set containing ni objects in class i. Running the classifier on this test set is the same as conducting k binomial experiments, where, in the ith experiment, the sample size is ni and the probability of success is simply Ai. Let gi be the number of correctly labeled objects belonging to class i in the test set. Then, conditional on the recall rate Ai, gi follows a binomial distribution: (29) (gi|Ai)∼Bin(ni,Ai)

The probability mass function of (gi|Ai=a) is thus(30) pgi|Ai(gi)=(nigi)agi(1−a)ni−gi    ∝agi(1−a)ni−gi

In the Bayesian setting, Eq. (28) is the prior and Eq. (30) is the likelihood. To get the posterior PDF, we simply multiply the prior with the likelihood: (31) fAi|g(a)∝fAi(a)×fgi|Ai(gi)   ∝a1−1(1−a)1−1×(1−a)ni−gi   =a1+gi−1(1−a)1+ni−gi−1

Thus, with respect to the binomial likelihood function, the Beta distribution is conjugate to itself. The posterior recall rate Ai also follows a Beta distribution, now with parameters (32) (Ai|gi)∼Beta(1+gi,1+ni−gi)

Our goal is to have a balanced accuracy rate, A, that puts an equal weight in each class. One way to achieve this is to take the average of the individual recalls: (33) A=1k∑i=1kAi =1kAT

Here we have defined AT to be the sum of the individual recalls. We call (A|g) the posterior balanced accuracy, where g = (g1, …, gk). Most of the time, we simply want to calculate its expected value: (34) E[A|g]=1k E[AT|g]   =1k∫a⋅fAT|g(a) da

Let us call this the MPBA. Note that there is no closed form solution for the PDF fAT|g(a). However assuming that AT is a sum of k independent Beta random variables, fAT|g(a) can be approximated by numerically convolving k Beta distributions. The independence assumption is reasonable here, since there should be little to no correlation between the individual recall rates. For example, knowing that a classifier is really good at recognizing stars does not tell us much about how well that classifier can recognize galaxies.

Having the knowledge of fA|g (a) will allow us to make violin plots, construct confidence intervals and do hypothesis tests. To get an expression for this, let us first rewrite the cumulative distribution function as(35) FA|g(a)=ℙ(A≤a|g)   =ℙ(1kAT≤a|g)   =ℙ(AT≤ka|g)   =FAT|g(ka)

Differentiating (Eq. (35)) with respect to a, we obtain the PDF of (A|g): (36) fA|g(a)=∂∂aFA|g(ka)   =∂∂a(ka)⋅∂∂kaFAT|g(ka)   =k⋅fAT|g(ka)

A Python implementation for the posterior balanced accuracy can be found on our GitHub repository (https://github.com/chengsoonong/mclass-sky).

Additional Information and Declarations

Competing Interests

Author Contributions

Data Availability

1 The best possible performance in each trial is obtained by the higher of: (1) the performance achieved by using all the labeled examples in the training set; and (2) the maximum value of the learning curves of all the methods.

The authors declare that they have no competing interests.

Alasdair Tran conceived and designed the experiments, performed the experiments, analyzed the data, prepared figures and/or tables, performed the computation work, authored or reviewed drafts of the paper, approved the final draft.

Cheng Soon Ong conceived and designed the experiments, authored or reviewed drafts of the paper, approved the final draft.

Christian Wolf authored or reviewed drafts of the paper, approved the final draft.

The following information was supplied regarding data availability:

The code of the experiments can be found at https://github.com/chengsoonong/mclass-sky.

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
