# Peer review of "Combining active learning suggestions"

_PeerJ Computer Science, doi:10.7717/peerj-cs.157_

## Round 0.1 · original submission · Major Revisions

Please follow all the reviewers' recommendations, and prepare a document explaining how have you modified the paper. It is specially relevant you consider extending the experimental part of the article, as well as make an appropriate statistical treatment of the results.

Reviewer 1 ·

Basic reporting

Generally speaking, the article is well written and pleasant to read.

A few comments on the structure and on the reporting:

- It might be more natural to place the paragraph on information-density weighting (lines 313-322) in Section 2.1.
- Using rank aggregation to combine multiple AL strategies has been proposed before [1], albeit in the context of active multi-task learning. I believe this should be acknowledged.
- The above reference can be found, e.g., in Burr Settles' book [2], which might also be a useful addition to the references.
- Table 4: "Max Performance" -> the metric is not specified.
- Figures 4 and 5: It would be helpful to indicate which quantiles are represented by the boxes and the whiskers.

[1] R. Reichart et al., Multi-Task Active Learning for Linguistic Annotations, ACL 2008.
[2] B. Settles, Active Learning, 2012, Morgan & Claypool

Experimental design

The research question that the authors seek to address is important and meaningful. Their approach---analyzing a large number of methods on a large number of datasets---is a priori interesting. The experimental methods are well described, and the code is easy to read.

However, I believe that there are important issues in the experimental design, such that it is not possible to gain a clear, objective understanding of the performance of the AL strategies. Hence, in its current state, the paper fails to properly address the research question.

1. The strength (eq. 25) is upper-bounded by 1, assuming that the discrete integrals are positive. However, looking at Fig. 4 and 5, we see a number of instances where the magnitude of the strength is much larger than 1, positively or negatively. This is due to the fact that $m(\max)$ in (25) is the performance achieved using all labeled data, which, in some cases, is *not* the best possible performance. In these cases, the discrete integral can be 0 or negative, and the strength might not even be well-defined. This seriously impacts the interpretability of the results. The following solutions might be explored:

- $m(\max)$ could be redefined as the best performance in the given run (such that by construction the discrete integrals will be positive), or
- multiple runs could be averaged before computing the strength, in order to decrease the variability in the performance, or
- the authors might rely less on strength measurements and more on the learning curves (Fig. 6 and 7), which should then be significantly improved, e.g., by averaging more runs (>=50) and by showing more strategies, or
- other performance measures might be considered, as they might prove to be less sensitive to the "nonpositive integral" problem.

2. There is no meaningful comparison with prior work. For example, I suggest considering the datasets & classifier of [3] and including the results of this paper in the plots, so that it is possible to get a clear comparison between the two.

3. The results seem to depend a lot on the choice of hyperparameters. These dependencies need to be explored more systematically. A few examples:

- Thompson sampling: the authors state that it favors certain heuristics a lot more than others. But this is likely going to be very different if $\tau^2_r$ and the initial $\mu_r$ are set differently.
- Poor performance of information-density weighted strategies: this obviously depends a lot on $\gamma$.
- Is there an effect of the size of the candidate pool?
- Regularization strength in Eq. 22: the choice of $C = 1000$ seems somewhat arbitrary and needs to be better explained.

[3] Y. Baram et al., Online Choice of Active Learning Algorithms, 2004, JMLR.

Validity of the findings

1. The authors claim that "with the small and medium-sized datasets, the effect of active-learning is minimal". Do the authors suggest that AL should only be used if one has access to a large unlabeled pool? Or that the number of AL iterations has to be sufficiently large? I believe this might be more of a measurement problem: with small datasets, there are fewer iterations, the test set is smaller, and the measured strength is noisier---in conclusion, this might just be an artefact of the experimental procedure. In any case. this should be clarified, perhaps by running additional experiences (e.g., by looking at a small sample of a large dataset, sample whose size matches that of the small datasets).

2. Line 353: "there is no statistical difference between the corresponding heuristic/combiner and passive learning". Clearly, looking e.g. at the median in Fig. 4., there seems to be a difference. Do the authors mean that the difference is not statistically significant?

Additional comments

Small comments:

- You refer multiple times to the "optimal heuristic". I believe "best heuristic" or "heuristic that works best" is more appropriate.
- Line 59: "test example" -> maybe "candidate example"?
- Eq. 5: why the $x$ in $r(x, h)$? The $x$ on the RHS is in a different scope (it is argmax'ed over). Maybe it should be $r(\mathcal{U}, h)$ instead.
- Sec 3.1. A general reference to multi-armed bandits would be welcome after the first sentence. Furthermore, I am not sure that the typical objective of MABs is to "minimize the number of pulls" (lines 132-133).
- Figure 1: why the dotted line? What is the difference with plain lines?
- Eq. 12: the way the function is defined implies that the bandit is able to see a reward for every arm, but that is not the case, I presume? I think the domain of the function should be revised.
- Eq 13: why $| v_r$ on the LHS?
- Algorithm 2, last line: "update the parameters of $B$" -> should be small case $b$.
- Line 201: "consider the KL-divergence" -> between which distributions?
- Line 203: "Algorithm" -> Algorithms
- Line 226: "we could only use one suggestion from one particular heuristic" -> from what I understand, this is not the case for EXP4.
- Line 236: $k(s(x; h))$ -> s(x:h) is a score (i.e., a real number), this is inconsistent with the definition of $k$ in Eq. 20.
- Line 256: Why is it necessary to rescale the ranks? I don't see why this would change the aggregation.
- Line 292: "with an L2 loss" -> I suspect you mean "with an L2 regularizer".
- Eq. 22: $w^T \theta$ should probably be $\theta^T \theta$.
- Line 316: "vetor" -> vector
- Line 393: "Thompson sampling requires" -> assumes
- Line 394 "This can be hard to satisfy" -> This assumption is unrealistic.

Reviewer 2 ·

Basic reporting

- The introduction should be improved, highlighting better the motivations and the main contributions of the work.

- Generally speaking, the writing of the paper is good.

- In the session “related work”, the authors should explain better the reasons to only analyze and describe the Uncertainty Sampling and Version Space Reduction strategies. The authors wrote that the other heuristics (query strategies) are very computationally expensive, but they should explain further this situation.

- “Out of the five heuristics discussed, how do we know which one is the most optimal one?” → This question is quite interesting, but I think that the author should consider to rewrite it. According to the Non-Free-lunch theorem, that can be applied to active learning too, there is not a method that performs the best for all possible problems. Maybe, the question should be focused to encounter a method with a “optimal” performance for a particular set of problems.

- “There have been some attempts to combine active learning suggestions in the literature. Baram et al. (2004) used the EXP4 multi-armed bandit algorithm to automate the selection process. Hsu and Lin (2015) studied an improved version, EXP4.P, along with importance weighting to estimate the rewards using only the training set. This paper empirically compares the following four algorithms: Thompson sampling, OC-UCB, kl-UCB, and EXP3++.”

The authors should not restrict the state-of-the-art description to the single-label classification problem. For instance, in the following reference was proposed a method to average several rankings and obtain a consensus for selecting the next query in the context of multi-label learning.

Oscar Reyes, Carlos Morell, Sebastián Ventura, Effective active learning strategy for multi-label learning, Neurocomputing, Volume 273, 2018, Pages 494-508.

- The authors should improve the explanation of the proposed method.

- Does the proposed method fulfill the Condorcet criterion?

Experimental design

- It is not explained how the authors obtained probabilities from the logistic regression base classifier.

- What were the criteria to select the 11 datasets used in the experimental study?. On the other hand, the authors should increase the number of datasets in order to make the study more reliable.

Validity of the findings

- My main concern about this work is that the authors assumed that only one sample is selected in every active learning iteration. However, nowadays this scenario is much less used since the selection of several samples at the same time is preferred in the majority of real-world applications. To select a set of queries in each active learning iteration you can follow either a myopic active learning strategy or a batch-mode active learning one. There are main differences between these two kinds of strategies. The authors should explain this situation in the paper, and portrait some tracks explaining how to adapt the proposed method to this type of scenarios.

- The discussion and conclusions are well stated, they are limited to the results obtained.

Additional comments

Generally speaking, I consider that this work is interesting, but it should be significantly improved before it could be considered for publication. I suggest that authors review the work attending the comments made by the reviewer.

---

## Round 0.2 · Minor Revisions

You only have to follow the recommendations of reviewer 1 in the new version of the paper.

Reviewer 2 ·

Basic reporting

The authors have addressed successfully the most of the comments raised by the reviewers. Two minor comments that should be addressed before publishing the paper.

1- “More sophisticated batch-mode active learning heuristics have been proposed, for example by using cluster analysis (Xu et al., 2007) or by maximizing the diversity in a batch (Brinker, 2003). How to aggregate suggestions from these heuristics is an interesting problem for future work.”

There are other interesting and recent batch-mode appoaches that the authors should consider to mention, for example the multi-objective approach proposed in:

Reyes, O., & Ventura, S. (2018). Evolutionary strategy to perform batch-mode active learning on multi-label data. ACM Transactions on Intelligent Systems and Technology (TIST), 9(4), 46.

2- The first reference has a lot of authors, maybe using the first author name followed by “et al” could be ok..

Experimental design

The experimental study is ok.

Validity of the findings

The description of the results and the discussion are ok.

---

## Round 0.3 · accepted · Accept

This version meets all the reviewers' expectations, and the paper is ready for publication. Congratulations!